# Review of the Neotropical Species of the *Elachista praelineata* Species Group (Lepidoptera, Elachistidae, Elachistinae) with Identification Keys and Description of a New Species from Bolivia

**DOI:** 10.3390/insects14010062

**Published:** 2023-01-09

**Authors:** Virginijus Sruoga, Jekaterina Havelka

**Affiliations:** Institute of Biosciences, Life Sciences Center, Vilnius University, Saulėtekio Ave. 7, LT-10257 Vilnius, Lithuania

**Keywords:** Microlepidoptera, mining moths, morphology, mitochondrial COI, DNA, South America, taxonomy

## Abstract

**Simple Summary:**

We studied small, dully coloured grass-miner moths (Elachistidae, Elachistinae). This is a large subfamily widely distributed throughout the world, but only a few species are known in the Neotropics. To facilitate further exploration, we reviewed one of nineteen species group within the largest genus, *Elachista*, along with identification keys of species. The *E. praelineata* species group now comprises of 37 described species, which are distributed in the Americas, Africa, Asia, and Australia. Neotropical species of the group are still underexplored, especially compared with that of Palearctic and Nearctic regions. So far only five species are known to belong to this group, mainly from the Ecuadorian Andes. In this study, we aimed to review and update information on the *E. praelineata* species group in the Neotropics. Here, we describe in detail a new species *Elachista stonisi* sp. nov. from western Bolivia. The female of *E. albisquamella* Zeller, 1877 is described for the first time and the female of *E. lata* Sruoga, 2010 is associated with the male. The description of new species and the association of females with males are supported morphologically and are confirmed by analysis of partial sequences of mitochondrial COI.

**Abstract:**

Neotropical species of the *Elachista praelineata* species group are reviewed. Five species are recognized in the Neotropics. A new species, *Elachista stonisi* sp. nov., and female of *E. albisquamella* Zeller, 1877 are described and illustrated with photographs of the adults, some of the immature stages, female genitalia, and leaf mines. The female of *E. lata* Sruoga, 2010 is associated with the male based on morphology and the comparison of partial mitochondrial COI sequences. Identification keys to the Neotropical species of *E. praelineata* species group, based on male and female genitalia, are provided.

## 1. Introduction

*Elachista* Treitschke, 1833 is the most species-rich, worldwide genus of the subfamily Elachistinae Bruand, 1850 (family Elachistidae Bruand, 1850). The moths are small with wingspans ranging from 4 to 20 mm [1]. Larvae are obligate leaf miners on monocots (almost solely grasses or sedges) [2,3]. The genus is represented by 14 described species in the Neotropical region [4]. The current concept of *Elachista* is based on the phylogenetic studies by Kaila [5] and Kaila and Sugisima [6]. The genus is now considered to comprise of four subgenera and 19 species groups [4].

The *E. praelineata* species group was established by Kaila [7,8]. Most members of the group are dark in appearance, with a pale transverse fascia near the middle of the forewing and a pair of pale spots on the distal part of the wing. Some species have also a small discal spot. The male genitalia are characterized mainly by prolonged cucullus, an indistinct hump at the distal fold of costa, and reduced valval process. A large tuft of hair-like scales in the female tergum 7 is characteristic [1]. The *E. praelineata* species group now comprises of 37 described species, which are distributed in Americas, Africa, Asia, and Australia (Table 1). Neotropical species of the group are still underexplored, especially compared with that of Palearctic and Nearctic regions. So far, only five species are known to belong to this group, mainly from the Ecuadorian Andes [9].

Recent collecting efforts by J. R. Stonis and A. Diškus in Bolivia have resulted in the discovery of two species belonging to the *E. praelineata* species group. One of them is new to science and one species is a new record to Bolivia.

This study was undertaken to review and update information on the species of *E. praelineata* species group in the Neotropics. The description of a new species is supported morphologically and is confirmed by the analysis of partial sequences of mitochondrial COI genes (318 bp or 685 bp fragments). The comparison of COI fragments was also used to clarify the taxonomic status of an earlier unnamed female (species VS307) from Ecuador [9] and to confirm the association of sexes of *E. albisquamella* Zeller, 1877. For the taxonomic keys, all known Neotropical species of the *E. praelineata* species group are included.

## 2. Materials and Methods

### 2.1. Morphological Study

Specimens examined in the present study were swept from low vegetation during the daytime and reared from leaf mines. A lectotype of *Elachista albisquamella* was examined at the Natural History Museum, London, UK (NHMUK).

Adult external morphology was examined with MBS-10 and Euromex Stereo Blue stereomicroscopes. The width of the head was measured between the inner edges of the antennal bases. Genitalia preparations followed standard techniques described by Robinson [42], adapted for the Elachistinae as described by Traugott-Olsen and Nielsen [2]. The female genitalia were stained with Chlorazol Black (Direct Black 38/Azo Black) and mounted on slides with Euparal. The photographs of the adults were taken with a Canon EOS 80D camera fitted with a MP-E 65 mm Canon macro lens, attached to a macro rail (MJKZZ Qool Rail). Images of the genitalia were taken with a Novex B microscope and a E3ISPM12000KPA digital camera. The morphological nomenclature follows Kaila [1,5] and Kristensen [43].

The descriptions of *Elachista stonisi* sp. nov. and *E. albisquamella* are based on the material deposited in the collection of the Museum für Naturkunde (MfN), Berlin, Germany. The other material mentioned in this study is housed in the Insect Collection of Life Sciences Center of Vilnius University, Vilnius, Lithuania (VU) and the Natural History Museum, London, UK (NHMUK).

### 2.2. DNA Extraction, Amplification, Sequencing and Data Analysis

Total genomic DNA was extracted from hind legs or thoraces removed from the dry adult specimens (Table 2) using a GeneJet Genomic DNA purification kit (Thermo Scientific). Samples were incubated with proteinase K overnight at 56 °C. For the amplification of partial COI sequences, the following primer pairs were used: LepF/MH-R1 and MF1/LepR [44]. PCR amplification was carried out in a thermal cycler (Eppendorf) in 25 μL of reaction mixture [45], containing 2 μL genomic DNA, 2.5 μL of 10 xTaq polymerase buffer with KCl, 2.5 μL of dNTP mix (2 mM), 2.5 μL of MgCl2 (2.5 mM), 2.5 μL of each primer (1 μM), 0.2 μL of Taq polymerase (recombinant, 5 u/μL) (Thermo Scientific), and nuclease-free water to 25 μL (Thermo Scientific). The cycling parameters were as follows: initial denaturation at 95 °C for 5 min (1 cycle), denaturation at 95 °C for 30 s, annealing at 45 °C for 30 s, extension at 72 °C for 30 s (40 cycles in total), and final extension at 72 °C for 5 min (1 cycle). PCR products were visualized on 1.5% agarose gel stained with ethidium bromide, purified using Gene Jet PCR purification kit (Thermo Scientific), and then sequenced at Macrogen Europe (Amsterdam, the Netherlands). The amplification primers were also used as sequencing primers.

DNA sequences for each specimen were confirmed with both sense and anti-sense strands and aligned in the BioEdit Sequence Alignment Editor [46]. If both fragments were amplified successfully for a particular sample, the sequences were concatenated. Otherwise, PCR products obtained with primers LepF and MH-R1 were used for further analysis. GenBank accession numbers and BOLD Process IDs are given in Table 2. Partial COI sequences were tested for stop codons, and none were found. To evaluate sequence diversity between samples, uncorrected p-distances were calculated, and a Neighbour-Joining tree was constructed with MEGA11 [47].

## 3. Results

### 3.1. Taxonomy

Key to the Neotropical species of the *Elachista praelineata* species group, based on male genitalia (male of *E. stonisi* sp. nov. is unknown and not included in the key).

*Valva straight, not* curved towards cucullus ([9]: Figure 24) ……………………………………………………………………………………………………………… *E. lata*

-Valva strongly curved and broadened towards cucullus …………………………………………………………………………………………………………………… 2

2.Sclerotized phallic tube straight ([8]: Figure 14B) ………………………………………………………………………………………………………………… *E. albisquamella*

-Sclerotized phallic tube curved …………………………………………………………………………………………………………………………………………………… 3

3.Sclerotized phallic tube gradually bent, s-shaped, apex blunt ([9]: Figures 11, 15 and 16) ……………………………………………………………………… *E. adunca*

-Sclerotized phallic tube curved at basal 1/2 and 4/5, apex tapered ([9]: Figures 18 and 21) ……………………………………………………………………………… *E. laxa*

Key to the Neotropical species of the *Elachista praelineata* species group, based on female genitalia (females of *E. adunca* and *E. laxa* are unknown and not included in the key).

Corpus bursae densely covered with distinct internal spines (this paper, Figure 1E) ……………………………………………………………………………… *E. albisquamella*

-Corpus bursae without internal spines …………………………………………………………………………………………………………………………………………………… 2

2.Ostium bursae occupying 1/2 width between apophyses anteriores; signum angulate ([9]: Figures 38 and 40) ………………………………………………………… *E. lata*

-Ostium bursae occupying 3/5 width between apophyses anteriores; signum straight (this paper, Figure 1D,E) ……………………………………………… *E. stonisi* sp. nov.


***Elachista adunca* Sruoga, 2010**


*Elachista adunca* Sruoga, 2010: 36 (Figures 5, 6, and 10–16, male) [9].

**Material examined.** 1 ♂ (holotype), Ecuador, Pichincha Province, 11 km NW Aloag, 24.II.2007, V. Sruoga leg.; gen. slide VS310 (VU); BOLD Process ID: EPRAE006-23.

**Diagnosis.***Elachista adunca* is a medium-sized and dark-coloured species. In wing pattern and male genitalia, this species is rather close to *E. albisquamella* Zeller, 1877, known from Colombia (for illustrations refer to [8]) and *E. laxa* Sruoga, 2010, known from Ecuador (for illustrations refer to [9]). The male of *E. adunca* differs by the absence of visible cilia in the flagellum of the antenna, while in *E. albisquamella* and *E. laxa* the flagellum is densely ciliated. The main differences between these three species are found in the morphology of the sclerotized phallic tube (see the key). In addition, the median ridge of the vinculum in *E. adunca* is narrow, while in *E. albisquamella* and *E. laxa* it is broad. The saccus in *E. adunca* is short and narrow, whereas in *E. laxa* it is short and wide.

**Biology.** Unknown.

**Distribution.** Known only from the type locality in northern Ecuador.

**Remarks.** The comparison of holotype partial COI sequences (318 bp) well support the original description of *E. adunca* based on morphological characters and its distinctness from morphologically similar *E. laxa* (7.23%) and *E. albisquamella* (5.97% from male or 6.29% from female) (Table 3). The differences of COI fragments between *E*. *adunca* and other Neotropical representatives of *E*. *prealineata* species group analysed in this study range from 11.32% to 11.95% (Table 3).


***Elachista albisquamella* Zeller, 1877**


*Elachista albisquamella* Zeller, 1877: 447 [10]; Kaila, 2000: 181 (Figure 14, male) [8].

(Figure 1A–E and Figure 2A–L)

**Figure 1 insects-14-00062-f001:**
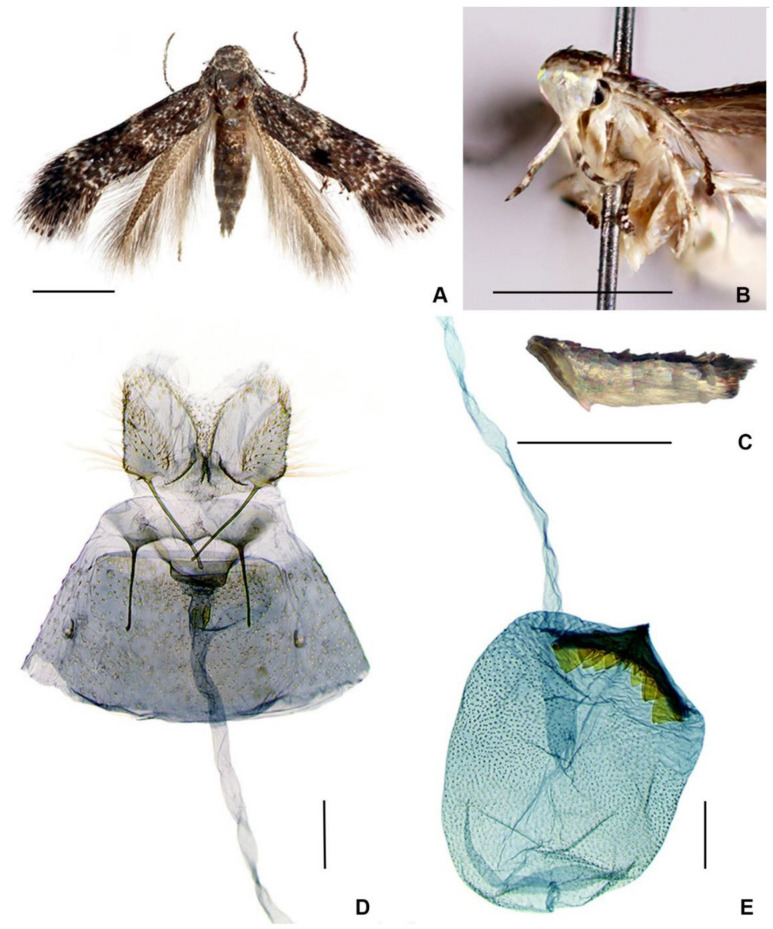
*Elachista albisquamella*, female: (**A**) adult; (**B**) head, fronto-lateral view; (**C**) abdomen, lateral view; (**D**) caudal part of genitalia; and (**E**) ductus and corpus bursae. Scale bars: (**A**–**C**) = 1 mm; (**D**,**E**) = 0.1 mm.

**Material examined.** 1 ♂ (lectotype), Colombia, Bogota; gen. slide 28538 (NHMUK); 8 ♂, 4 ♀, Bolivia, Yungas Province, Coroico (Cascada), mining larvae 10.VI.2018, ex pupae 14.VI.–5.VII.2018, J. R. Stonis & A. Diškus leg.; gen. slides VS476, VS474 (MfN); BOLD process IDs: EPRAE007-23 ♂, EPRAE008-23 ♀.

**Diagnosis.***Elachista albisquamella* is a small and dark-coloured species with indistinct wing markings. In wing pattern and male genitalia, this species is rather close to *E. adunca* Sruoga, 2010 and *E. laxa* Sruoga, 2010, known from Ecuador (for illustrations refer to [9]). The male of *E. albisquamella* differs by the length of cilia in the flagellum of the antenna which is about 1/3 of diameter of the shaft, while in *E. laxa* it is 2/3 of the diameter of the shaft and in *E. adunca* the flagellum is without visible cilia. In male genitalia *E. albisquamella* can be distinguished most easily by the sclerotized phallic tube being straight and longer than the valva, whereas in *E. adunca* and *E. laxa* it is variously curved and shorter than the valva. Female genitalia differ from other Neotropical species of *E. praelineata* species group with known females by having a corpus bursae with dense internal spines, large triangularly-shaped teeth, and by a dorsal wall with few very small spines.

**Description of female** (Figure 1A–C). Forewing length 2.75–3.25 mm; wingspan 6.0–7.1 mm (*n* = 4). Head: frons—shining white; vertex and neck tuft—greyish brown, mottled by paler bases of scales; labial palpus—2 times as long as the width of the head, slightly recurved in apical part, shining white above except the apex of the second segment and base, middle and apex of the third segment—greyish brown; antenna—greyish brown, basal articles are weakly annulated with paler rings, and the underside—greyish white in basal 1/3. Thorax and tegula—greyish brown, slightly mottled with white. Forewing: ground colour formed from basally whitish and distally greyish brown scales making a mottled appearance; indistinct transverse whitish fascia at 1/3 of the wing; a small spot consisting of raised brownish black scales on a fold just beyond the fascia, slightly below the fold; indistinct whitish spot at 2/3 length of costa and a similar one on the dorsum just before it; fringe scales—brownish grey, and fringe line—blackish brown. Hindwing—grey-brown, with fringe concolorous. Abdomen—above greyish brown, white below; the tergum 7 with small, greyish brown tuft of hair-like scales.

**Female genitalia** (Figure 1D,E). The papilla analis is longer than it is wide, distally rounded, with small setose swelling latero-basally. The apophysis posterioris is longer than the apophysis anterioris. The ostium bursae are situated in the intersegmental membrane between sternites seven and eight, occupying about 1/2 width between apophyses anteriores, and a dorsal wall covered with tiny spines. The antrum is wide and very short, weakly sclerotized, and abruptly tapered towards short colliculum. This colliculum is sclerotized anteriorly, leaving a short membranous zone between the sclerotization and the antrum, and the length of the colliculum is about 1/2 length of the apophysis anterioris. The ductus bursae is almost equally wide for the whole length, about 7.5 times longer than the apophysis anterioris, without internal spines. The corpus bursae is rounded, and densely covered with distinct internal spines; The signum is large, bluntly boomerang-shaped, and with stout triangular teeth.

**Biology.** The mines in Bolivia were found in the leaves of unidentified bamboo growing at an altitude of about 1700 m (Figure 2J–L). The egg is laid singularly on the upper side of the leaf, as an elongated oval, flattened dorsolaterally, showing up as a white translucent ‘shell’ at start of the mine, apparently with rather coarsely sculptured chorion (Figure 2D). The mine begins as a narrow elongate track (Figure 2A–C) which enlarges to an elongated, large irregular blotch. Frass is deposited in two irregular lines in the narrow part of the mine (Figure 2E); in the widened part, frass is arranged along margins of the mine leaving an unfilled central part of the blotch. The final length of the mine is about 35–40 mm, occasionally up to 50 mm. The full-grown larva pupates on the leaf in a flimsy cocoon (Figure 2F–I). The cocoon varies in density and orientation of silk filaments covering the pupa. The pupa is highly modified; a thorax and abdomen with lateral ridges produced into short, curved spines and with dark brown dorsolateral lines.

**Distribution.** Colombia [8,10]; Bolivia, Yungas Province, 16°13′18″ S, 67°41′33″ W (new record).

**Flight period.** Probably two generations per year. Moths in Colombia were captured in mid-February and mid-March [10]. In Bolivia, adults occur from mid-June to early July.

**Remarks.** The female paralectotype of *E. albisquamella* was designated by Kaila [8], however, its genitalia were not examined. The association of sexes here is based on simultaneous finding of females’ and males’ larvae mining the same host plant in the same sample. Additionally, the association of sexes is supported by molecular data. COI fragments of male and female analysed in this study (685 bp) differed by 3 substitutions (0.44%) (Table 3). The differences of COI fragments between *E*. *albisquamella* and other Neotropical representatives of the *E*. *prealineata* species group analysed in this study range from 5.97% with *E*. *adunca* to 11.68% with *E*. *stonisi* sp. nov. (Table 3).


***Elachista lata* Sruoga, 2010**


*Elachista lata* Sruoga, 2010: 39 (Figures 22–31, male; 32–34, 37–40, female) [9].

**Material examined.** 1 ♂ (holotype), Ecuador, Tungurahua Province, Banos env., 10.II.2007, V. Sruoga leg.; gen. slide VS475 (VU); 1 ♀, same label as holotype; gen. slide VS307 (VU); BOLD Process IDs: EPRAE002-23 ♀, EPRAE003-23 and EPRAE004-23 ♂.

**Diagnosis.***Elachista lata* is a small and dark-coloured species with inconspicuous white wing markings. Among all Neoptropical species of the *E. praelineata* species group, *E. lata* is the only one with a straight valva, a large oval spinose knob of gnathos, and a very short digitate process.

**Biology.** Unknown.

**Distribution.** Only known from type locality in central Ecuador.

**Flight period.** Based upon the two specimens available, adults fly in early February.

**Remarks.** Prior to this study, *E. lata* was known as male only. This species was originally described from the Tungurahua Province of Ecuador on the basis of the male holotype [9]. Besides, Sruoga [9] described a female as species VS307 (not named) from the same locality, belonging to the same species group, while noting that it could be conspecific with *E. lata*. Externally, the female of VS307 slightly differs from male of *E. lata* by more expanded white markings of the forewing. However, the partial COI sequences (318 bp) of the holotype of *E. lata* and of the single known specimen of VS307 differ by only one base substitution (0.31%) (Table 3). Therefore, these two specimens are here considered as conspecific. The differences of COI fragments between *E*. *lata* and other Neotropical representatives of the *E*. *prealineata* species group analysed in this study range from 9.43% with *E*. *stonisi* sp. nov. to 12.26% with *E*. *laxa* (Table 3).


***Elachista laxa* Sruoga, 2010**


*Elachista laxa* Sruoga, 2010: 36 (Figures 7–9, 17–21, male) [9].

**Material examined.** 1 ♂ (holotype), Ecuador, Pichincha Province, 11 km NW Aloag, 24.II.2007, V. Sruoga leg.; gen. slide VS312 (VU); BOLD Process ID: EPRAE005-23.

**Diagnosis.***Elachista laxa* is a medium-sized and dark-coloured species with inconspicuous wing markings. In wing pattern and male genitalia, this species is rather close to *E. albisquamella* Zeller, 1877, known from Colombia and Bolivia (for illustrations refer to [8]) and *E. lata* Sruoga, 2010, known from Ecuador (for illustrations refer to [9]). The male of *E. laxa* differs by the length of cilia in the flagellum of the antenna, which is about 2/3 the of diameter of the shaft, while in *E. albisquamella* it is about 1/3 of the diameter of the shaft, and in *E. adunca* the flagellum is without visible cilia. The main differences between these three species are found in the morphology of the sclerotized phallic tube (see the key).

**Biology.** Unknown.

**Distribution.** Only known from type locality in northern Ecuador.

**Flight period.** The only one specimen was captured in late February.

**Remarks.** The comparison of holotype partial COI sequences (318 bp) well support the original description of *E. laxa* based on morphological characters and its distinctness from the morphologically similar *E. adunca* (7.23%) and *E. albisquamella* (8.18%) (Table 3). The differences of COI fragments between *E*. *laxa* and other Neotropical representatives of the *E*. *prealineata* species group analysed in this study range from 11.95% with *E*. *lata* to 12.58% with *E*. *stonisi* sp. nov. (Table 3).


***Elachista stonisi* sp. nov.**


(Figure 3A–E and Figure 4A–G)

urn:lsid:zoobank.org:act:0D2E35DC-2131-4B63-9545-CDE3DC96DF80

**Material examined.** Holotype: ♀ Bolivia, Yungas Province, W. Coroico, Yolosa, Death Road, mining larvae 16.VI.2018, ex pupa 5–8.VII.2018, J. R. Stonis & A. Diškus leg.; gen. slide VS475 (MfN). Paratypes: 3 ♀, same label as holotype (MfN); BOLD Process ID: EPRAE001-23.

**Diagnosis**. In wing pattern and female genitalia, this species closely resembles *Elachista lata* Sruoga, 2010, known from Ecuador (for external characters and female genitalia refer to [9]). However, the forewing in the new species is without a white patch basally and with a white discal spot. The main differences in female genitalia between *E. stonisi* and *E. lata* are (1) the ostium bursae in *E. stonisi* occupying 3/5 of the width between the apophyses anteriores, in *E. lata* it is about 1/2; (2) the signum in *E. stonisi* is straight, whereas in *E. lata* it is angulate.

**Description of female** (Figure 3A–C). Forewing length 3.1–3.4 mm; wingspan 6.8–7.3 mm (*n* = 4). Head: Frons and vertex—whitish grey; neck tuft—grey-brown; the labial palpus is 1.2 times as long as the width of head, whitish grey above, and grey-brown below; antenna—grey-brown, and weakly annulated with paler rings basally. Thorax and tegula—grey-brown. Forewing: ground colour—grey-brown; white transverse fascia from the costa to the dorsal margin before the middle; a triangular white spot at 5/7 of costa, with another similar spot just before it in the dorsal margin; small discal spot elongated, white, at 4/5 of wing length in the middle; all white markings with weak metallic lustre; fringe scales—brownish grey, broad fringe scales along the termen basally—greyish white and distally—blackish brown, forming a fringe line. Hindwing—brownish grey, its fringe scales are somewhat paler. The abdomen is grey-brown above, and whitish below; the tergum 7 has very large and dense, brownish grey tufts of hair-like scales.

**Female genitalia** (Figure 3D,E). The papilla analis is longer than it is wide, distally rounded. The Tergum 7 reniform and is 1.5 times longer than it is wide. The apophysis posterioris is slightly longer than the apophysis anterioris and curved in the apical part. The ostium bursae is situated in the intersegmental membrane between sternites seven and eight, occupying 3/5 of the width between the apophyses anteriores, and the dorsal wall is strongly spinosed. The antrum is cup-shaped, inwardly spinosed, weakly sclerotized, and abruptly tapered towards the short colliculum. The colliculum is sclerotized, anteriorly leaving a short membranous zone between the sclerotization and antrum, the length of the colliculum being about 1/4 the length of the apophysis anterioris. The ductus bursae and corpus bursae are without internal spines; the signum is straight and narrow, with short teeth.

**Biology.** The mines were found in the leaves of unidentified *Poaceae* growing at an altitude of about 1390 m (Figure 4G). The mine begins as a narrow elongate track (Figure 4 A–D) which enlarges to an elongated, large irregular blotch. Frass is deposited in slender lines in the narrow part of the mine; in the widened part, frass is arranged irregularly. Pupation takes place on the leaf, where it is attached to the surface by cremaster and a silken girdle. The pupa has a dorsal ridge and a pair of more distinct dorso-lateral ridges on the abdomen (Figure 4E,F).

**Distribution.** So far, this species is known only from western Bolivia, Yungas Province, 16°13′8″ S, 67°44′54″ W.

**Flight period.** Based on the specimens available, adults fly in July.

**Etymology.** The new species is named in honour of well-known Lithuanian entomologists, Professor Jonas Rimantas Stonis.

**Remarks.** The comparison of paratype partial COI sequences (685 bp) well supports the description of *E. stonisi* sp. nov. based on morphological characters and its distinctness from the morphologically similar *E. lata* (9.43% from male or 9.75% from female) (Table 3). The differences of COI fragments between *E*. *stonisi* sp. nov. and other Neotropical representatives of the *E*. *prealineata* species group analysed in this study range from 11.39% with *E*. *albisquamella* to 12.58% with *E*. *laxa* (Table 3).

### 3.2. COI Fragment Analysis

Partial COI sequences were obtained for all known Neotropical species of the *Elachista praelineata* species group (Table 2). In this study, 685 bp COI fragment of *E*. *albisquamella* and *E*. *stonisi* sp. nov., and 318 bp partial COI sequences of *E*. *adunca*, *E*. *lata* and *E*. *laxa*, were analysed. Pairwise, uncorrected p-distances between samples are given in Table 3. The lowest values were between samples representing the same species: 0.00–0.31% for *E*. *lata* and 0.43% for *E*. *albisquamella*. The distances between samples representing different species ranged from 5.97% between *E*. *adunca* and *E*. *albisquamella*, to 12.58% between *E*. *laxa* and *E*. *stonisi* sp. nov. (Table 3).

Neotropical species of the *E. praelineata* species group formed two major clusters in a Neighbour-Joining tree based on partial COI sequences (Figure 5). One cluster consists of sequences belonging to two species, *E. lata* and *E. stonisi* sp. nov., and the other one is made up of sequences representing samples of *E. albisquamella*, *E. laxa*, and *E. adunca*. Sequence distances corresponded well with morphological data: morphologically similar species showed lower values of uncorrected p-distances.

## 4. Discussion

The *Elachista praelineata* species group is distributed in all major biogeographical regions worldwide, including the Neotropics. With the description of *E. stonisi* sp. nov., the *E. praelineata* species group now has 5 species from the Neotropical region. The majority of species, i.e., 4, are known only from the type localities in Ecuador. However, their status as endemics will be confirmed when more extensive regional studies have been conducted. Only *E. albisquamella* has a wider range, as it is now known to occur in Colombia and Bolivia.

The biological knowledge of the *E. praelineata* species group is scarce. The host plants are indicated for only 40% of species worldwide [4], and none of them feed on bamboo. The host plants of Neotropical species of the *E. praelineata* species group were completely unknown prior to this study. Here we report *E. albisquamella* larvae feeding on bamboo. In the Neotropics, the diversity of bamboo is huge, with hundreds of species [48]. If species of *E. praelineata* group specializing in them, even marginally match this diversity, one may assume that a vast diversity of elachistines is awaiting discovery.

The Nearctic region, while vast and with a hugely diverse biota, is virtually unexplored with respect to the collection and study of the *E. praelineata* species group. The number of species undoubtedly is expected to be much higher. The main reason for our limited understanding of this group of moths in the Nearctic region is a lack of adequate fieldwork. A great amount of work still needs to be done—from collecting, to the description of new taxa and the natural history of species.

## Figures and Tables

**Figure 2 insects-14-00062-f002:**
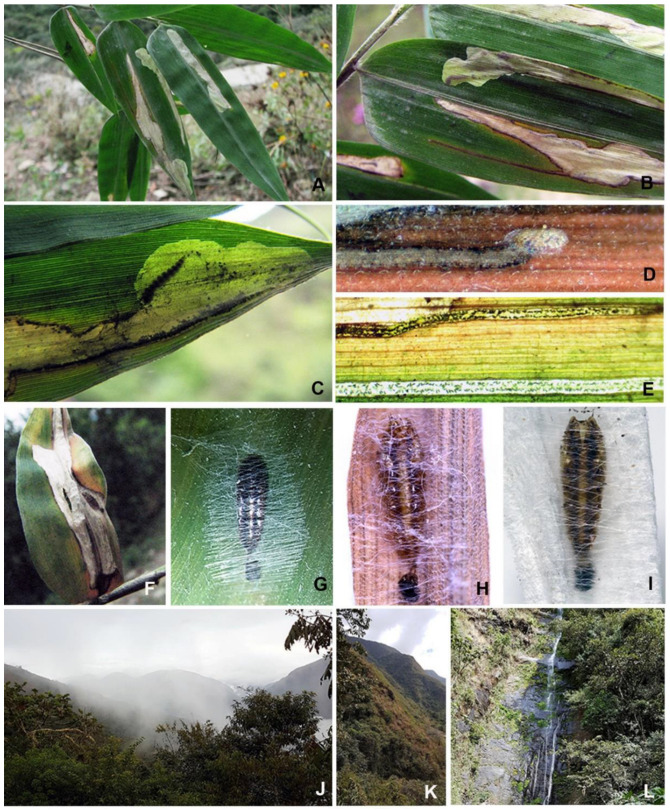
Immature stages and habitat of *Elachista albisquamella*: (**A**–**C**) leaf-mines; (**D**) egg; (**E**) frass in the mine; (**F**–**I**) pupa; and (**J**–**L**) habitat where the leaf mines were found, Yungas Province, Bolivia.

**Figure 3 insects-14-00062-f003:**
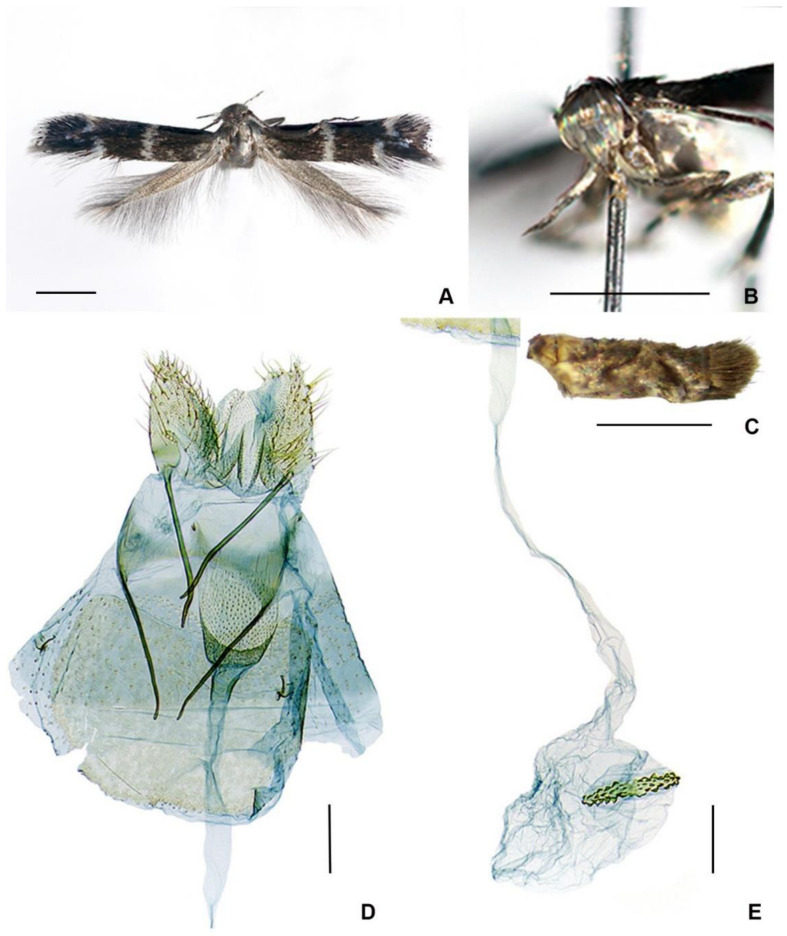
*Elachista stonisi* sp. nov., female, holotype: (**A**) adult; (**B**) head, fronto-lateral view; (**C**) abdomen, lateral view; (**D**) caudal part of genitalia; and (**E**) ductus and corpus bursae. Scale bars: (**A**–**C**) = 1 mm; (**D**,**E**) = 0.1 mm.

**Figure 4 insects-14-00062-f004:**
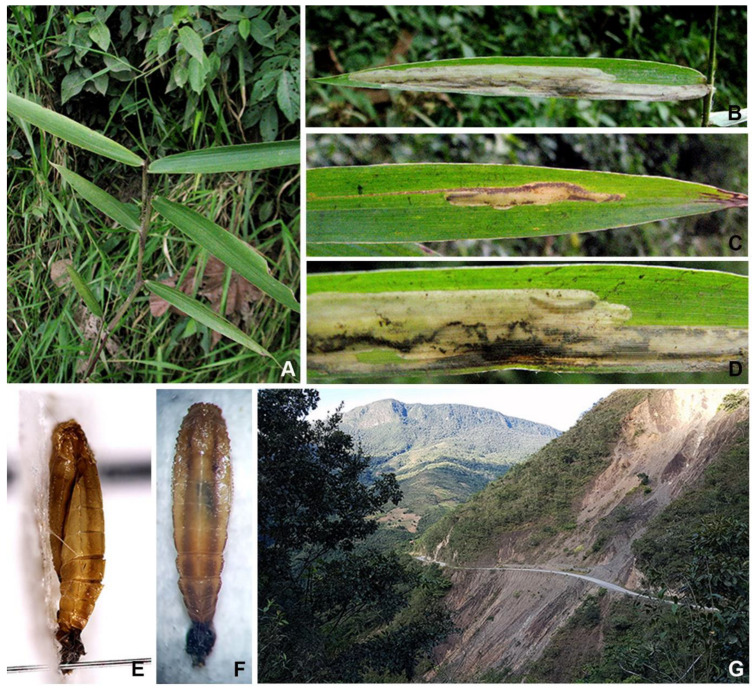
Immature stages and habitat of *Elachista stonisi* sp. nov.: (**A**) host plant, *Poaceae* sp.; (**B**–**D**) leaf-mines; (**E**,**F**) pupa; (**G**) habitat where the leaf mines were found, Yungas Province, Bolivia.

**Figure 5 insects-14-00062-f005:**
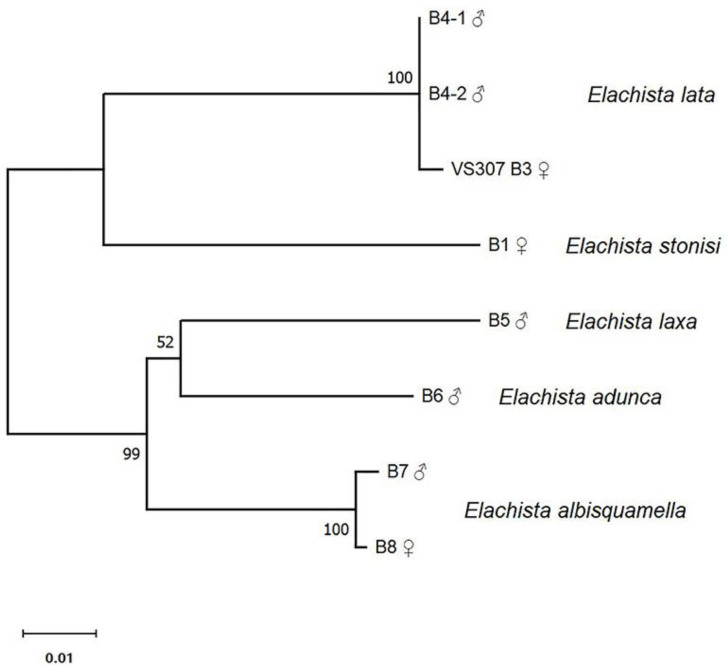
A Neighbour-Joining tree showing relationships among partial sequences of mitochondrial cytochrome oxidase subunit I (COI) in samples of 5 species of Neotropical species of *Elachista praelineata* species group. The percentage of replicate trees in which the associated samples clustered together in the bootstrap test (1000 replicates) are shown next to the branches.

**Table 1 insects-14-00062-t001:** Species, distribution, and host plants of the *Elachista praelineata* species group.

*Elachista* Species	Host Plant	Distribution	References
Neotropical Region			
*adunca* Sruoga, 2010	Unknown	Ecuador	[9]
*albisquamella* Zeller, 1877	*Bambusoideae* sp.	Colombia; Bolivia	[8,10], present study
*lata* Sruoga, 2010	Unknown	Ecuador	[9]
*laxa* Sruoga, 2010	Unknown	Ecuador	[9]
*stonisi* sp. nov.	*Poaceae* sp.	Bolivia	Present study
*Nearctic* Region			
*aranella* Kaila, 1999	Unknown	Canada	[7]
*aristoteliella* Kaila, 1999	Unknown	USA	[7]
*eilinella* Kaila, 1999	Unknown	USA	[7]
*guilinella* Kaila, 1999	Unknown	USA	[7]
*ibunella* Kaila, 1999	Unknown	USA	[7]
*miriella* Kaila, 1999	Unknown	USA	[7]
*nienorella* Kaila, 1999	Unknown	USA	[7]
*praelineata* Braun, 1915	*Hystrix patula* Moench	USA	[7,11,12,13]
*radiantella* Braun, 1922	*Panicum clandestinum* (L.), *P. dichotomum* (L.)	USA	[7,12,14]
*serindella* Kaila, 1999	Unknown	USA	[7]
*solitaria* Braun, 1922	*Panicum* sp.	USA	[7,12,14]
*staintonella* Chambers, 1878	Unknown	USA	[7,12,15]
*turinella* Kaila, 1999	Unknown	USA	[7]
*Palearctic* Region			
*amamii* Parenti, 1983	*Digitaria timorensis* (Kunth) Balansa, *D. adscendens* (H. B. K.) Henr., *Thuerea involuta* (Forst.) R. Br.	Japan; Taiwan; Thailand	[16,17,18]
*bromella* Chrétien, 1915	*Bromus madritensis* L.	Algeria; Morocco	[4,19,20]
*caliginosa* Parenti, 1983	*Eccoilopus cotulifer* A. Camus, *Spodiopogon sibiricus* Trin.	Russia; Japan	[16,17,21]
*canariella* Nielsen & Traugott-Olsen, 1987	*Bromus* sp.	Canary Islands	[22,23]
*encumeadae* Kaila & Karsholt, 2002	*Festuca donax* Lowe	Madeira	[24]
*fasciocaliginosa* Sugisima, 2005	*Miscanthus sinensis* Anderss.	Japan	[17]
*kurokoi* Parenti, 1983	*Oplismenus undatifolius* (Ard.) P. Beauv.	Japan; Korea; Nepal	[16,17,25,26]
*minor* Sruoga, 2022	Unknown	Nepal	[26]
*miscanthi* Parenti, 1983	*Miscanthus sinensis* Anderss.	Japan; Taiwan	[16,17]
*nepalensis* Traugott-Olsen, 1999	Unknown	Nepal	[27]
*sicula* Parenti, 1978	*Briza maxima* L., *Bromus erectus* Hudson, *Lagurus ovatus* L.	Italy	[28,29]
*simulans* Sruoga, 2022	Unknown	Nepal	[26]
*tabghaella* Amsel, 1935	Unknown	Israel	[30,31]
Afrotropical Region			
*kakamegensis* Sruoga & De Prins, 2009	Unknown	Kenya	[32]
*merimnaea* Meyrick, 1920	Unknown	South Africa	[33,34]
*semophanta* Meyrick, 1914	Unknown	Malawi	[35,36]
*trifasciata* (Wollaston, 1879)	Unknown	Saint-Helena	[37,38,39]
*Oriental* Region			
*brachyplectra* Meyrick, 1921	Unknown	Sri Lanka; Indonesia	[40,41]
Australasian Region			
*aurita* Kaila, 2011	*Oplismenus* sp.	Australia	[1]

**Table 2 insects-14-00062-t002:** Specimens used for molecular analysis of Neotropical species of *Elachista praelineata* species group.

Species	DNA Isolate	GenBank Accession/BOLD Process ID/Length (bp)
*Elachista stonisi*; female; paratype	B1	OP955854/EPRAE001-23/685
*Elachista* sp. VS307; female	B3	OP955855/EPRAE002-23/318
*Elachista lata*; male; holotype	B4-1	OP955856/EPRAE003-23/318
*Elachista lata*; male; holotype	B4-2	OP955857/EPRAE004-23/318
*Elachista laxa*; male; holotype	B5	OP955858/EPRAE005-23/318
*Elachista adunca*; male; holotype	B6	OP955859/EPRAE006-23/318
*Elachista albisquamella*; male	B7	OP955860/EPRAE007-23/685
*Elachista albisquamella*; female	B8	OP955861/EPRAE008-23/685

**Table 3 insects-14-00062-t003:** Uncorrected pairwise p-distances (%) between partial sequences of mitochondrial cytochrome oxidase subunit I (COI) from samples of 6 species of Neotropical species of *Elachista praelineata* species group.

	B1	B3	B4-1	B4-2	B5	B6	B7	B8
*E*. *stonisi*; B1								
*Elachista* sp. VS307; B3	9.75							
*E*. *lata*; B4-1	9.43	0.31						
*E*. *lata*; B4-2	9.43	0.31	0.00					
*E*. *laxa*; B5	12.58	12.26	11.95	11.95				
*E*. *adunca*; B6	11.95	11.64	11.32	11.32	7.23			
*E*. *albisquamella*; B7	11.68	11.01	10.69	10.69	8.18	5.97		
*E*. *albisquamella*; B8	11.39	10.69	10.38	10.38	8.18	6.29	0.44	

## Data Availability

All data are contained within the article.

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
