# Peer review of "Review of the Neotropical Species of the Elachista praelineata Species Group (Lepidoptera, Elachistidae, Elachistinae) with Identification Keys and Description of a New Species from Bolivia"

_insects, 2023, doi:10.3390/insects14010062_

Round 1

Reviewer 1 Report

This paper is a significant contribution to the knowledge of Neotropical microlepidoptera

1.     Line 96. Is the DNA sequence within the barcode fragment? If yes to facilitate the access to the reader to the molecular data and their associated metadata (collecting data, photos of vouchers) I strongly recommend authors to submit their data to the barcoding data base BOLD (https://www.boldsystems.org/). You have already the sequences in Genbank but the advantage of having the data in BOLD is that all the metadata (collecting data, photo of the voucher) is better associated to the sequence. When submitting the sequences and specimen data fiel to BOLD please let BOLD staff know that your sequences are already in genbank so they can make the link between BODL and genebank.

2.     When submitting your data to BOLD, you will have to assign unique codes (sample Ids) to each barcoded voucher. Please add those sample IDs to Table 2 and also in the “material examined” section in particular the holotype. Please add those sample IDs as printed labels to all the vouchers sequenced.

3.     Please add a section on taxa sampling to material and methods explaining the methods used to collect the material used in this study (ie. erared, light trap, netted by day….)

4.     Table 1. Please add the known host plant data for each species

5.     Legend figure 5, please indicate what the values below or above branches mean (bootstrap values?).

6.     Figure 5: the node between stonisi and lata does not have a bootstrap value, is that normal?

7.     Please comment about the host plant range of each species

Author Response

Thank you very much for the reviewing. Here is a point-by-point response to the reviewers’ comments:

1) Yes, the DNA sequence is within the barcode fragment. We submitted data to BOLD.

2) BOLD IDs are added in Table 2. 

3) We added information on taxa sampling to material and methods.

4) The known host plants were added to Table 1.

5) The legend is corrected.

6) Yes, we think it is normal. This is the original tree produced by MEGA11.

7) The host plants are known only for two here discussed species (Elachista albisquamella, E. stonisi), we commented this for each other three species discussed here.

Reviewer 2 Report

Well done manuscript completely ready for publication

Author Response

Thank you very much for the reviewing. 

Reviewer 3 Report

This is an unusualy well prepared manuscript, almost without errors. 

I have only two minor qestions about the text (see manuscript)

I can recommend its publication in Insects!

Author Response

Thank you very much for the reviewing. 

Here is a response to the reviewers’ comments in the manuscript:

Piont 1. Should it be BIB instead of clade?

Response 1. We are not sure we understand the meaning of BIB. We changed “clade” to “cluster”.

Point 2. Why partial when it is 685 bp?

Response 2. The full length of COI gene is about 1500 bp, e.g. in Drosophila yakuba (for details see Clary and Wolstenholme (1985), The mitochondrial DNA molecule of Drosophila yakuba: nucleotide sequence, gene organization and genetic code. Journal of Molecular Evolution, 22: 252-271.). Therefore, we refer COI sequences analyzed in our study as “partial”.